# Machine Learning Methods for the Prediction of the Inclusion Content of Clean Steel Fabricated by Electric Arc Furnace and Rolling

**Estela Ruiz** [1], **Diego Ferreño** [2,*], **Miguel Cuartas** [3], **Lara Lloret** [4], **Pablo M. Ruiz del Árbol** [4], **Ana López** [1], **Francesc Esteve** [1] and **Federico Gutiérrez-Solana** [2]

1 Global Steel Wire, Nueva Montaña s/n, 39011 Santander, Spain; estela.ruiz@globalsteelwire.com (E.R.); ALOPEZD@globalsteelwire.com (A.L.); francesc.esteve@globalsteelwire.com (F.E.)
2 LADICIM (Laboratory of Science and Engineering of Materials Division), E.T.S. de Ingenieros de Caminos, Canales y Puertos, University of Cantabria, Av. Los Castros 44, 39005 Santander, Spain; gsolana@unican.es
3 GTI (Group of Information Technologies), E.T.S. de Ingenieros de Caminos, Canales y Puertos, University of Cantabria, Av. Los Castros 44, 39005 Santander, Spain; miguel.cuartas@unican.es
4 IFCA (Instituto de Física de Cantabria), University of Cantabria—CSIC, Av. Los Castros s/n, 39005 Santander, Spain; lloret@ifca.unican.es (L.L.); pablo.martinez@unican.es (P.M.R.d.Á.)
* Correspondence: ferrenod@unican.es

**Abstract:** Machine Learning classification models have been trained and validated from a dataset (73 features and 13,616 instances) including experimental information of a clean cold forming steel fabricated by electric arc furnace and hot rolling. A classification model was developed to identify inclusion contents above the median. The following algorithms were implemented: Logistic Regression, K-Nearest Neighbors, Decision Tree, Random Forests, AdaBoost, Gradient Boosting, Support Vector Classifier and Artificial Neural Networks. Random Forest displayed the best results overall and was selected for the subsequent analyses. The Permutation Importance method was used to identify the variables that influence the inclusion cleanliness and the impact of these variables was determined by means of Partial Dependence Plots. The influence of the final diameter of the coil has been interpreted considering the changes induced by the process of hot rolling in the distribution of inclusions. Several variables related to the secondary metallurgy and tundish operations have been identified and interpreted in metallurgical terms. In addition, the inspection area during the microscopic examination of the samples also appears to influence the inclusion content. Recommendations have been established for the sampling process and for the manufacturing conditions to optimize the inclusionary cleanliness of the steel.

**Keywords:** inclusion content; machine learning; classification; random forest

## 1. Introduction

The steelmaking industry imposes tight controls on steel cleanliness because non-metallic inclusions (NMIs) negatively influence both the manufacture and the application of steel products. NMIs of different nature (mostly oxides, sulfides and nitrides) are always present in steel, but their amount and size greatly varies. They come from the combination between the low solubility metallic elements present in the liquid steel with elements such as oxygen, sulfur or nitrogen. The type, size, shape and quantity of NMIs depend on the steel grade and the details of the steelmaking and casting processes. NMIs are classified as "endogenous" or "exogenous". The former occurs within the liquid steel, precipitating out during cooling and solidification (for example, during deoxidation, because of the intentional addition of calcium to combine with sulfur). Exogenous inclusions are, in turn, entrapments of materials from refractory interfaces, slag or other materials in contact with the melt. Endogenous inclusions are typically more uniformly distributed than exogenous

ones. The current understanding of the origin and classification of NMIs has been thoroughly reviewed by Vasconcellos [1], emphasizing the interplay between thermodynamics, steel and slag chemical composition as well as the melt shop processing characteristics.

The detrimental consequences of NMIs on the performance of steel in structural and mechanical applications has been extensively described in the literature [2], showing the influence on properties such as fatigue [3,4], fracture strength [5], ductility [5] or corrosion [2]. In addition, NMIs can lead to problems during fabrication (for example, they can clump together and clog the nozzles in casting), welding, machinability as well as surface quality degradation, among others.

The density of NMIs is lower than that of the liquid metal; therefore, the majority of NMIs are incorporated to the floating slag facilitating their removal and improving the cleanliness of the steel. Inclusion absorption by slag occurs in three stages [6]: (i) transport of the inclusion to the interface between steel and slag, (ii) movement of the inclusion to the interface, breaking the surface tension of the steel and (iii) incorporation of the NMI into the slag. The probability of elimination increases with the size of the NMIs; in this sense, stirring promotes the coalescence of NMIs and favors their elimination.

The inclusionary content in clean steels is frequently determined through the microscopic examination of polished surfaces of hot rolled product, as in the methods proposed by the standards ASTM E45 18a [7], EN 10247: 2017 [8], DIN 50602 [9] and ISO 4967: 2013 [10]. According to DIN 50602 [9], metallographic samples are observed under the optical microscope and rated using the comparative pictures of inclusions provided by the standard; as a result, the sample is assigned an index that measures the inclusionary cleanliness and allows different heats to be compared. This morphological classification is expected to correspond to the chemical-based grouping, as the shape of a particle is strongly related to the composition [1,5]. DIN 50602 [9] provides a procedure in which NMIs are assigned to a category based on similarities in morphology (size, shape, concentration and distribution). An index that measures the inclusion content can be calculated separately for the oxide and sulfide components or as a total value. In the so-called "method K", NMIs are counted and weighted according to their area, starting from a specified size of inclusion upwards. This index indicates the content of such inclusions in the product. Specifically, DIN 50602 [9] provides a set of diagrams constructed line by line on the basis of a 2" geometrical series for the area of NMIs, containing forms of inclusion typical for steel, the inclusion area doubling from one diagram to the next in each column. Inclusions of equal area but differing in length × width or frequency, are shown on the same line next to the basic column for each type of inclusion. The following types of inclusion are distinguished: sulfide inclusions of elongated type, oxide inclusions of fragmented type, (aluminum oxides), oxide inclusions of elongated type (silicates) and oxide inclusions of globular type.

The formation of NMIs in steel is an extraordinarily complex phenomenon controlled by specific thermodynamic aspects and by the melt shop processing variables. The term "inclusion engineering" has been coined to define the processing conditions that are beneficial or, at least, harmless, for the final steel product. This includes the amount, size, distribution, chemistry and mechanical properties of the NMIs generated [2]. In summary, inclusions can be purposefully tailored to promote desirable properties in steel. The thermodynamic fundamentals of NMI formation are well established [11–13] but there is no agreement about the thermodynamic data required for specific processes such as the generation of alumina inclusions or in the case of elements with limited solubility in iron, such as Mg and Ca [1]. With the development of computational thermodynamics, it has been possible to deal with problems that were unsolvable through conventional procedures [14–16], however, the lack of thermodynamic data remains a problem.

Computational thermodynamics is just but one of the fields where inclusion engineering is benefiting from the use of computers. Several contributions can be mentioned where numerical procedures were employed to understand the very specific processes behind the formation of NMIs. Pfeiler et al. [17] modeled the 3D turbulence flow of the steel

melt and the trajectories of NMIs comparing the one-way coupling (which considers only the impact of the melt flow on the trajectories of the dispersed phases) and the two-way coupling, obtaining better results with this second approach. Choudhary and Ghosh [18] developed a computational procedure for the prediction of the composition of NMIs in various solid fractions: the predictions of the model were compared with data coming from the literature as well as with compositions determined in cast billet samples using SEM-EDS, obtaining good agreement. In recent times, Machine Learning (ML) algorithms are being increasingly implemented as a helpful tool to improve the accuracy in determining the inclusionary content of steels. For instance, computer vision, which relies on ML classification algorithms, has been employed to determine whether a feature on a Scanning Electron Microscope (SEM) image was an inclusion or not [19]. Classification was conducted by means of convolutional neural networks (CNNs) obtaining an accuracy of 98% and enormous gains in time-saving. In addition, clustering ML algorithms have been used [20] to automatically (i.e., without implementing user-defined rules) group inclusions with similar chemical compositions and to find physical clusters of inclusions (i.e., groups of smaller inclusion particles that have joined together).

This paper is focused on developing an ML model for the reliable prediction of the K3 index, as defined by the DIN 50602 [9] standard, of a clean cold forming steel fabricated by electric arc furnace and hot rolling. To the best of these authors' knowledge, no previous study has addressed this issue. The processing parameters (inputs or features) and the target variable (K3 index) were collected in the context of the quality control program of the Global Steel Wire (GSW, Spain) company. The remainder of this paper is organized as follows: Section 2 (Material and Methods) describes: (i) the properties of the steel characterized in the study, (ii) the procedure to obtain the K3 index following the standard DIN 50602 [9] and (iii) the ML algorithms used for the correlation of the input and output parameters. The experimental results are presented and examined in Section 3, discussion is in Section 4 and, finally, the main conclusions drawn are compiled in Section 5.

## 2. Materials and Methods

### 2.1. Cold Heading Steel: Properties and Fabrication

The term cold-formed/heading steel refers to steel products shaped by cold-working carried out near room temperature, such as rolling, pressing, stamping, bending, etc. In this research, wire rods for cold heading were characterized to obtain the K3 index following the DIN 50602 [9] standard. Rods were hot-rolled from as cast 180 mm × 180 mm billets with class D surface quality, according to the EN 10221 standard [21]. The appropriate hardenability of the product after quenching was ensured by means of Jominy tests [22]. Three categories were included in the study, namely, aluminum killed, boron and chromium alloyed steels.

The fabrication of cold forming steel comprises the following four major stages: electric arc furnace (EAF), ladle furnace (LF), continuous casting (CC) and hot rolling (HR). These are briefly described hereafter [23–25]:

- In the EAF, steel scrap, direct reduced iron and hot briquetted iron are melted by means of high-current electric arcs to obtain liquid steel with the required chemistry and temperature. Lime and dolomite are included in the EAF to promote the formation of slag, which favors the refining of steel and reduces heat losses. Molten steel is poured into the transportation ladle where ferroalloys and additives are added to form a new slag layer.
- The so-called secondary metallurgy occurs in the LF; there, the final chemical composition and the temperature of the steel are adjusted. Deoxidizers, slag formers and other alloying agents are added for the refining. Molten steel is stirred by means of a stream of argon to homogenize the temperature and composition and to promote the flotation of NMIs within the slag. The chemistry of steel and slag, different temperatures and the amounts of fluxes and argon injected are monitored in the LF stage.

- During CC, liquid steel is poured from the ladle into the tundish (a small distributer that controls the flow rates and feeds the mold) provoking the solidification of steel in the form of billets. Chemical compositions and temperatures make up the parameters recorded at this stage.
- Rods are obtained from billets through HR. The steel is passed through several pairs of rolls to reduce the thickness, the final cross-section being typically between 5 and 30 mm in diameter. To facilitate the process, the temperature of steel during forming is above the recrystallization temperature. Rods are coiled after HR.

The total number of attributes collected throughout the whole fabrication process of the rods is 73. These attributes will be the features of the ML analysis.

### 2.2. The K-Index as a Measure of the Degree of Purity

The standard DIN 50602 [20] describes the examination of special steels for NMIs of sulfidic and oxidic nature. NMIs are assigned to a category based on similarities in morphology (size, shape, concentration and distribution). The procedure is calibrated for microscopic inclusions, which are defined as those with a projected area less than 0.03 mm$^2$. The content of NMIs in the form of sulfides and oxides defines the degree of purity, which can be measured following the method K. In this method, all NMIs from a specified size upwards are recorded and the degree of purity of a cast or a batch is expressed through the K-index, which is defined as the sum obtained by counting inclusions weighted by a factor that increases with their area; the result is normalized to an area of 1000 mm$^2$. According to the standard, in general, the degree of purity of a cast or a batch must be obtained from at least six specimens (to deal with the intrinsic statistical scatter of the distribution of NMIs) whose polished surface will be, if possible, parallel to the direction of forming (e.g., rolling).

Four types of NMIs are distinguished by the standard [9], namely, elongated sulfide inclusions (SS), fragmented aluminum oxides (OA), elongated silicates (OS) and oxide inclusions of globular type (OG). Each of these types of NMIs are rated using the reference plates provided by the standard that enable the identification of the diagram in the plate that corresponds to the field under observation in the microscope (using a magnification of 100×). The K-index is calculated by adding the values for the oxides and sulfides.

The standard DIN 50602 [20] enables rating numbers between 0 and 8 to be selected by the user. "0" corresponds to the smallest microscopic inclusion that can be evaluated at a magnification of ×100; moreover, the size of the smallest NMI considered for the analysis increases with the rating number (the area of the smallest inclusions follows a $2^n$ geometric series, where n is the rating number). This study is entirely based on the K3 index.

### 2.3. Machine Learning Methods

The ML models have been developed and evaluated in the Python 3 programming language using libraries such as Numpy, Pandas, Scikit-Learn, Matplotlib and Seaborn, among others. The workflow of this ML project is summarized [26] in the following sections.

#### 2.3.1. Scope of the Analysis

The objective of a supervised learning model is to predict the correct label for new input data. Supervised learning can be split into two subcategories: regression and classification. The target in regression is a continuous number (for example, the K3 index in this research). In the first attempt, this study was addressed as a regression; however, in view of the meager results achieved, it was decided to treat it using a classification approach. A classification algorithm assigns a discrete (typically binary) class/category to the given data. Here, classes 0 and 1 were assigned to the datum points depending on whether their K3 index was below or above the median of the distribution, respectively. Therefore, this approach will make it possible to distinguish between "low" and "high" impurity contents in the steel (which are separated by the median).

### 2.3.2. Collecting Data

The dataset used in this study included 13,616 instances, i.e., steel coils that had been characterized to obtain the K3 index, and 73 features (see Section 2.1). These data were obtained in the context of the quality program of the company GSW.

### 2.3.3. Data Preprocessing

The ability to learn from ML models and the useful information that can be derived may be extremely influenced by data preprocessing [27]. This consists of cleaning the raw data so that they can be used to train the model. The 80/20 rule is widely accepted in ML: 80% of the time is spent in preprocessing while the remaining 20% is devoted to proper ML analysis. Preprocessing includes the following stages [28,29]:

- Outliers and meaningless values were removed from the dataset. Data outliers can mislead the training process resulting in longer training times and less accurate models. z-score is a common procedure to detect and remove outliers. The z-score indicates how many standard deviations a data point is from the sample's mean. In this research, outliers were defined as data points beyond $|z| > 3.0$.

- Multicollinearity is potentially harmful for the performance of the model and may reduce its statistical significance. More importantly, it makes it difficult to determine the importance of a feature to the target variable. The Pearson's correlation matrix of the dataset was obtained and one of the features of every couple with a correlation coefficient exceeding (in absolute value) 0.60 was removed (this selection was supported with engineering judgement). The final number of features was 37. Figure 1 shows the heatmaps of the original (left) and final (right) correlation matrices showing the reduction in the number of features as well as the removal of highly correlated ones (these matrices only include the numeric features).

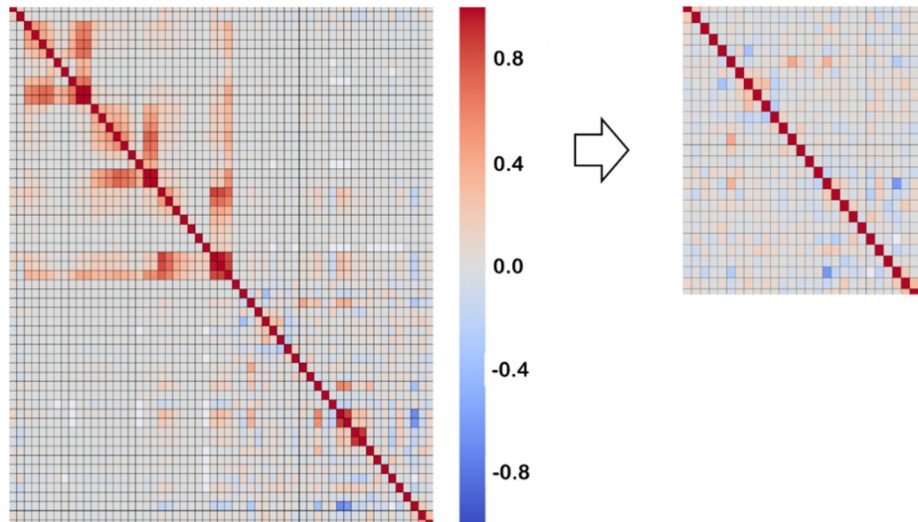

**Figure 1.** Heatmaps of the Pearson's correlation matrices on the original (**left**) and final (**right**) datasets. The number of features was reduced from 73 to 37. Notice that the regions in the left heatmap displaying large correlations are absent in the right heatmap.

- Standardization/feature scaling of a dataset is mandatory for some ML algorithms and advisable for others. Some regressors or classifiers, such as K-Nearest Neighbors or Support Vector Machines (see Section 2.3.4), calculate the distance in the feature's hyperspace between instances; this distance is governed by the features with the broadest values. For this reason, the range of all features must be normalized so that each one contributes approximately proportionately to the final distance. For other algorithms, such as Multi-Layer Perceptron (see Section 2.3.4), scaling is not compulsory but recommended because gradient descent converges much faster with

feature scaling. In this study, features were scaled though the StandardScaler provided by Scikit-Learn [30,31] which standardizes the features by removing the mean and scaling to unit variance.

- Missing data: Imputation is the process of replacing the missing values of the dataset by an educated guess. To avoid removing rows or columns, imputation was carried out by means of the KNNImputer provided by Scikit-Learn [32] in which missing values are estimated using the K-Nearest Neighbors algorithm (see Section 2.3.4).
- Handling categorical variables: Ordinal categorical variables were transformed using the Scikit-Learn LabelEncoder [33] and nominal categorical variables were subjected to the Scikit-Learn OneHotEncoder [34].

### 2.3.4. ML Algorithms

By virtue of the "No Free Lunch theorem" [35] there is not, a priori, an ML algorithm that works best for every problem. Therefore, the following models were employed in this study [28,29]:

- Logistic Regression (LR) [36] is considered as a baseline algorithm for binary classification. LR measures the relationship between the dependent variable and the independent variables using the sigmoid/logistic function (which is determined through a Maximum Likelihood Method). The logistic function returns the probability of every observation to belong to class 1. This real value in the interval (0, 1) is transformed into either 0 or 1 using a threshold value.
- In K-Nearest Neighbors (KNN) [37], classification or regression is conducted for a new observation by summarizing the output variable of the "K" closest observations (the neighbors) with weights that can be either uniform or proportional to the inverse of the distance from the query point. The simplest method to determine the closeness to neighbor instances is to use the Euclidean distance. The performance of KNN may fail in problems with a large number of input variables (curse of dimensionality).
- Support Vector Machine (SVM) was originally designed as a classifier [38] but may also be used for regression and feature selection. In classification, SVM determines the optimal separating hyperplane between linearly separable classes maximizing the margin, which is defined as the distance between the hyperplane and the closest points on both sides (classes). Many datasets are highly nonlinear but can be linearly separated after being nonlinearly mapped into a higher dimensional space [39]. This mapping gives rise to the kernel, which is selected by the user in a trial and error procedure on the test set.
- A Decision Tree (DT) is a non-parametric supervised learning method used for classification and regression [40]. Classification and Regression Trees were introduced in 1984 by Breiman et al. [41]. It is a flowchart-like structure in which each internal node represents a split based on a feature. The split with the highest information gain will be taken as the first split and the process will continue until all children nodes are pure, or until there is no information gain. "Gini" and "entropy" are common metrics to decide the best split. Leaf nodes (final nodes) represent the class labels. The main advantage of DTs is that the resulting model can easily be visualized and understood by non-experts. In addition, the DT may provide the feature importance, which is a score between 0 and 1 for each feature to rate how important each feature is for the decision a tree makes. The main downside is that they tend to overfit and provide poor generalization performance. In such cases, pruning methods can be implemented to control the complexity of Decision Trees.
- Ensemble algorithms [42] combine multiple "weak classifiers" into a single "strong classifier". A weak classifier is a classifier that performs slightly better than random guessing. Ensemble algorithms are classified into bagging-based and boosting-based, which are respectively designed to reduce variance and bias. Random Forest (RF) is a widely used bagging algorithm based on classification trees (weak learner). In RFs, each tree in the ensemble is built from a sample drawn with replacement (i.e., a

bootstrap sample) from the training set. In addition, instead of using all the features, a random subset of features is selected, further randomizing the tree. AdaBoost (AB), which stands for adaptive boosting, is the most widely used form of boosting algorithm. In this case, weak learners are trained sequentially, each one trying to correct its predecessor. In AB, the weak learners are usually Decision Trees with a single split, called decision stumps. Gradient Boosting (GB) is another ensemble algorithm, very similar to AB, which works by adding predictors sequentially to a set, each correcting its predecessor.

- An Artificial Neural Network (ANN) [43] contains a large number of neurons/nodes arranged in layers. A Multi-Layer Perceptron (MLP) contains one or more hidden layers (apart from one input and one output layers). The nodes of consecutive layers are connected and these connections have weights associated with them. In a feedforward network, the information moves in one direction from the input nodes, through the hidden nodes to the output nodes. The output of every neuron is obtained by applying an activation function to the linear combination of inputs (weights) to the neuron; sigmoid, tanh and Rectified Linear Unit (ReLu) are the most widely used activation functions. MLPs are trained through the backpropagation algorithm. Gradient descent, Newton, conjugate gradient and Levenberg–Marquardt are different algorithms used to train an ANN.

### 2.3.5. Training and Testing the Model on Data

The dataset comprises 13,616 instances and (after feature selection, see Section 2.3.3) 37 features. A total of 25% of the instances (3404 observations) were randomly extracted to form a test dataset that was later used to provide an unbiased evaluation of the models. This way, the model's performance is evaluated on a new set of data that were not seen during the training phase. This approach helps in avoiding overfitting (in this case, the algorithm learns the noise of the training set but fails to predict on the unseen test data). The inconvenience of a train/test split is that the results can depend on the particular random choice of the test set. To avoid this, Scikit-Learn [31,44] was used to implement 3-fold cross-validation on the 75% of remaining instances (10,212 observations) in order to select the best models and to optimize their hyperparameters through training and validation, avoiding overfitting. Model selection and hyperparameter optimization were conducted with GridSearchCV [31,45].

### 2.3.6. Evaluation Scores for Classification

The Confusion Matrix is often used to report the outcome of a classification analysis. It is a table with two rows and two columns that reports the predicted and the actual instances, providing the number of False Positives (*FP*s), False Negatives (*FN*s), True Positives (*TP*s) and True Negatives (*TN*s). *Accuracy*, *Recall*, *Precision* and *F1* are common scores in classification, see Equations (1)–(4) [28,29]:

$$Accuracy = \frac{TP + TN}{TP + FP + TN + FN} \tag{1}$$

$$Recall = \frac{TP}{TP + FN} \tag{2}$$

$$Precision = \frac{TP}{TP + FP} \tag{3}$$

$$F1 = 2\frac{Precision \cdot Recall}{Precision + Recall} \tag{4}$$

While *Recall* expresses the ability to find all relevant instances in a dataset, *Precision* expresses the proportion of relevant data points the model predicted to be relevant. Low *Recall* indicates many *FN*s and low *Precision* indicates a large number of *FP*s. The True and

False Positive Rates (*TPR*, which is equal to *Recall*, and *FPR*, also known as Specificity), see Equations (5) and (6) are also widely employed as scores for classification.

$$TPR = \frac{TP}{TP + FN} \tag{5}$$

$$FPR = \frac{FP}{FP + TN} \tag{6}$$

The Average Precision (AP) is defined as the integral below the curve that represents the *Precision* as a function of the *Recall*. The Receiver Operating Characteristic (ROC) curve [28,29] is a curve that relates the *TPR* on the *y*-axis versus the *FPR* on the *x*-axis. For a given classification model, adjusting the threshold makes it possible to move along the curve (reducing the threshold means moving to the right and upwards along the curve). The diagonal curve between points (0, 0) and (1, 1) corresponds to random guessing. The ROC curve results can be synthesized numerically by calculating the total Area Under the Curve (AUC), a metric which falls between 0 and 1 with a higher number indicating better classification performance. For a random classifier AUC = 0.5 while, for a perfect one, AUC = 1.0.

### 2.3.7. Relevance of Features: Feature Importance and Permutation Importance

Interpretability of ML models is in many instances as important as their prediction performance. One of the techniques to interpret a model is to identify which are the most important features. Feature Importance (FI) and Permutation Importance (PI) are two independent methods to assess the relevance of the features involved. Both techniques are implemented in Scikit-Learn [46,47]. FI is available for ensembles of trees. In a DT, the more a feature decreases the impurity, the more important the feature is. In RFs, the impurity decrease from each feature can be averaged across the trees of the ensemble to determine the final importance of the variable [48]. For classification, the measure of impurity is either the Gini impurity or the information gain/entropy while, for regression, it is the variance. In GB, the importance of a feature is calculated as the fraction of samples that will traverse a node that is split based on that variable. Then, the average across all trees is calculated to define the FI of the feature. PI is defined as the decrease in a model's score when a single feature value is randomly shuffled [41]. A number of advantages have been pointed out in favor of PI [46,49]. First, it is model-agnostic, i.e., it can be obtained from any model, not necessarily an ensemble of trees. Moreover, it has recently been observed that impurity-based FI can inflate the importance of numerical features and that categorical variables with a large number of categories are preferred [50].

### 2.3.8. Partial Dependence Plots

A Partial Dependence Plot (PDP) shows how a feature affects predictions or, more specifically, the marginal effect one feature has on the predicted outcome of a ML model. In a PDP the *x*-axis represents the values of the feature, while the *y*-axis displays the partial dependence [51] which, for binary classification, is the probability of the positive class.

## 3. Results

### 3.1. Description of the Distribution of the K3 Index

Figure 2 shows the histogram of the K3 index (after standardization) obtained from the 13,616 instances (coils) considered in the study. As can be seen, the distribution is positively skewed and the majority of the datum points are concentrated below the mean (which corresponds to the 67th percentile).

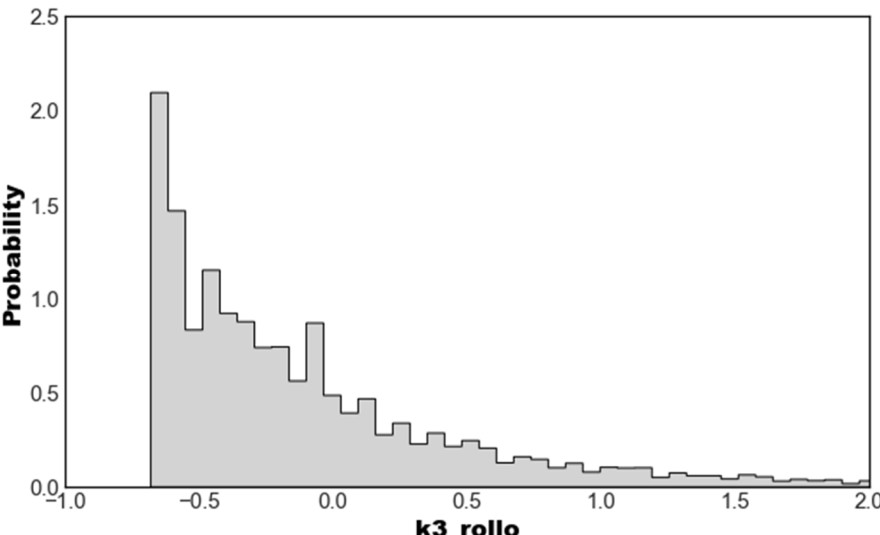

**Figure 2.** Histogram showing the distribution of the K3 index of the dataset available for the study.

### 3.2. Classification of Observations

This section is organized as follows: the performances of the ML algorithms previously described in Section 2.3.4 are compared in Section 3.2.1 to select the most reliable model. The selected algorithm will be used in Section 3.2.2 to assess the importance of the features and to identify the most relevant ones. Finally, the PDPs collected in Section 3.2.3 will enable the impact of each of the features to be expressed in quantitative terms.

### 3.2.1. Comparison of the Performance of the ML Models

Table 1 shows the scores obtained in the test dataset (25% of the instances, randomly selected), with each of the algorithms implemented. RF displays the best results surpassing the rest of the algorithms in six out of the seven scores. Accordingly, RF has been selected in this study as the optimal algorithm for the rest of the analyses. The ROC and *Precision–Recall* curves are represented for the test dataset in Figure 3. In addition, the confusion matrix, using a threshold of 0.5, is shown in Table 2. The following hyperparameters were obtained after cross-validation: n_estimators = 1000, max_depth = 20, max_features = 5, min_samples_split = 20, bootstrap = True.

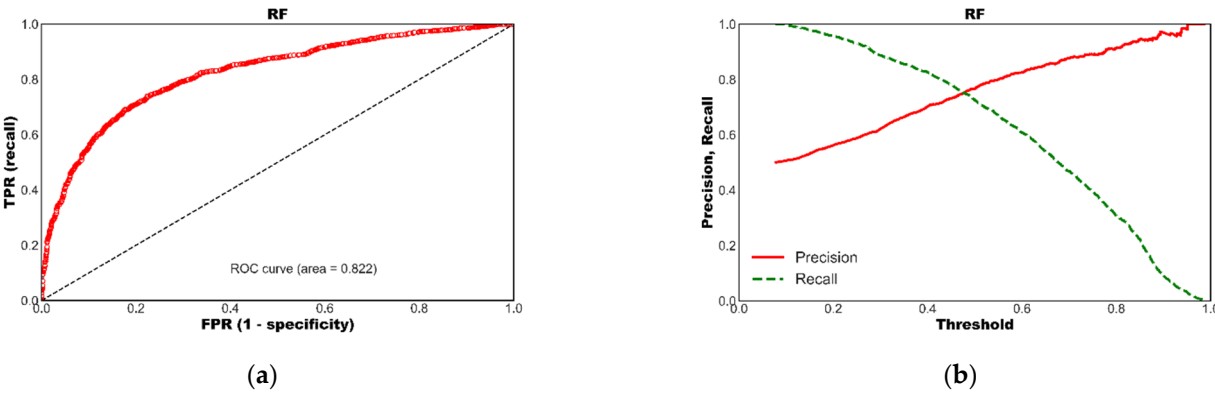

                (**a**)                                                 (**b**)

**Figure 3.** ROC (**a**) and *Precision–Recall* (**b**) curves obtained with the optimal RF.

**Table 1.** Summary of the scores obtained on the test dataset with the different classification algorithms to detect observations higher than the median. *Precision* and *Recall* were obtained for a threshold of 0.5.

|  | LR | KNN | DT | RF | AB | GB | SVC | MLP |
|---|---|---|---|---|---|---|---|---|
| AUC | 0.655 | 0.797 | 0.742 | 0.822 | 0.742 | 0.807 | 0.769 | 0.767 |
| *Accuracy* | 0.619 | 0.737 | 0.691 | 0.747 | 0.680 | 0.743 | 0.710 | 0.712 |
| *Precision* | 0.610 | 0.735 | 0.741 | 0.749 | 0.669 | 0.742 | 0.719 | 0.707 |
| *Recall* | 0.568 | 0.706 | 0.540 | 0.709 | 0.658 | 0.712 | 0.648 | 0.681 |
| *F1* | 0.588 | 0.720 | 0.626 | 0.728 | 0.663 | 0.726 | 0.682 | 0.694 |
| AP | 0.634 | 0.772 | 0.709 | 0.820 | 0.736 | 0.804 | 0.751 | 0.752 |

**Table 2.** Confusion matrix obtained on the test dataset with a threshold of 0.5 through the RF model optimized to detect observations higher than the median.

|  | Predicted | |
|---|---|---|
| **Actual** | **False** | **True** |
| False | 1332 | 373 |
| True | 466 | 1222 |

The pair of histograms shown in Figure 4 has been composed to provide a graphical representation of the discrimination power of the model. To elaborate this figure, the 500 instances with the highest and lowest values of the K3 index have been respectively selected, and the probability of belonging to each of the classes has been obtained using the corresponding RF model. These probabilities have been represented in the form of histograms. As can be seen, a clear separation is achieved, obtaining only a small overlap region between the distributions. This representation can be considered as another quality score of the models as well as a guarantee of the algorithm's ability to identify the underlying patterns in the dataset.

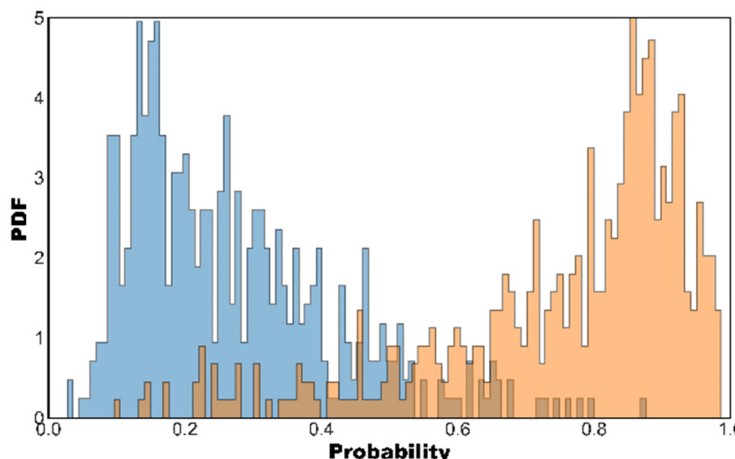

**Figure 4.** The histograms represent the probability provided by the optimized RF model of the 500 observations of the test set with the highest value of the K3 index and of the 500 observations with the lowest value of the K3 index, respectively, of belonging to category "1".

### 3.2.2. Importance of the Features

A first step to interpret the outcome of the ML analysis is to find out which features display a larger influence on the K3 index; this will help in implementing corrective measures in the fabrication process to improve the inclusionary cleanliness of the steel. As mentioned in Section 2.3.7, the FI approach suffers from certain limitations compared to the PI method. For this reason, this analysis is based on the latter. Another advantage of PI

is that the algorithm can be executed several times to estimate the uncertainty of the results. Results are represented in boxplots in Figure 5 for each of the features. The meaning and interpretation of these results are presented in Section 4.

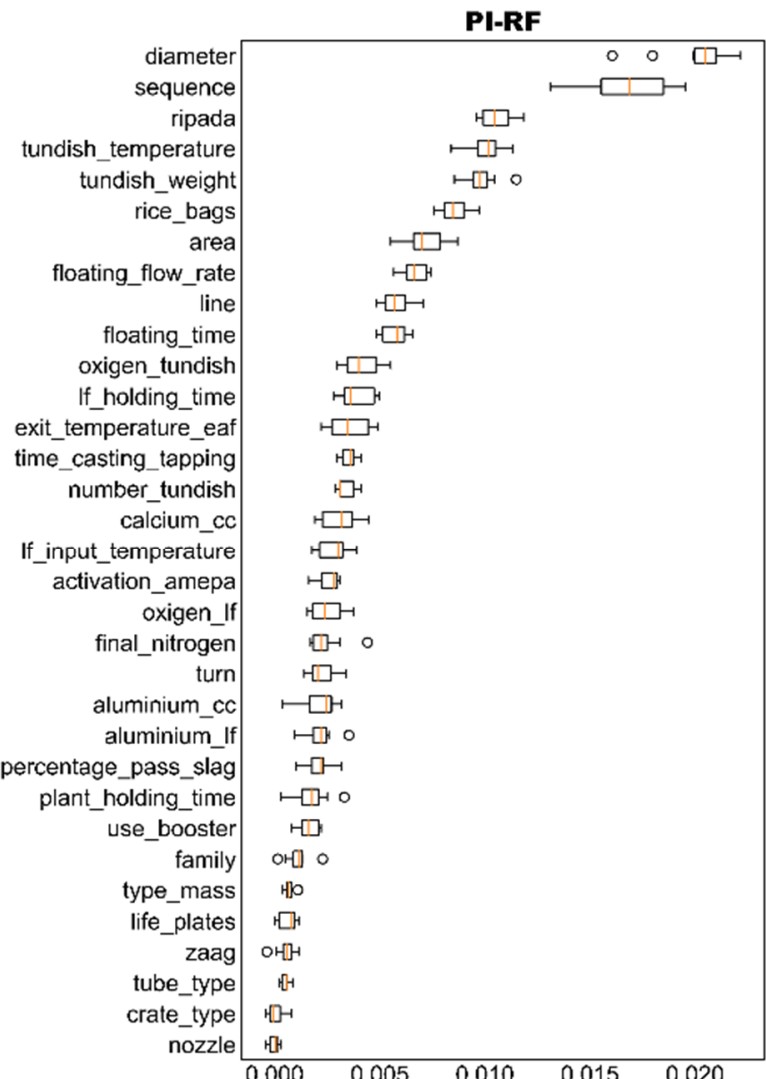

**Figure 5.** Boxplots showing the PI of the features through the RF optimized model.

### 3.2.3. Partial Dependence Plots

To complete the interpretation of the influence of each processing variable on the K3 index, it is necessary to obtain an estimate of the functional relationship between them. This is precisely the purpose of the PDPs, defined in Section 2.3.8. Figure 6 shows the PDPs obtained through RF of the seven most relevant features, see Figure 5.

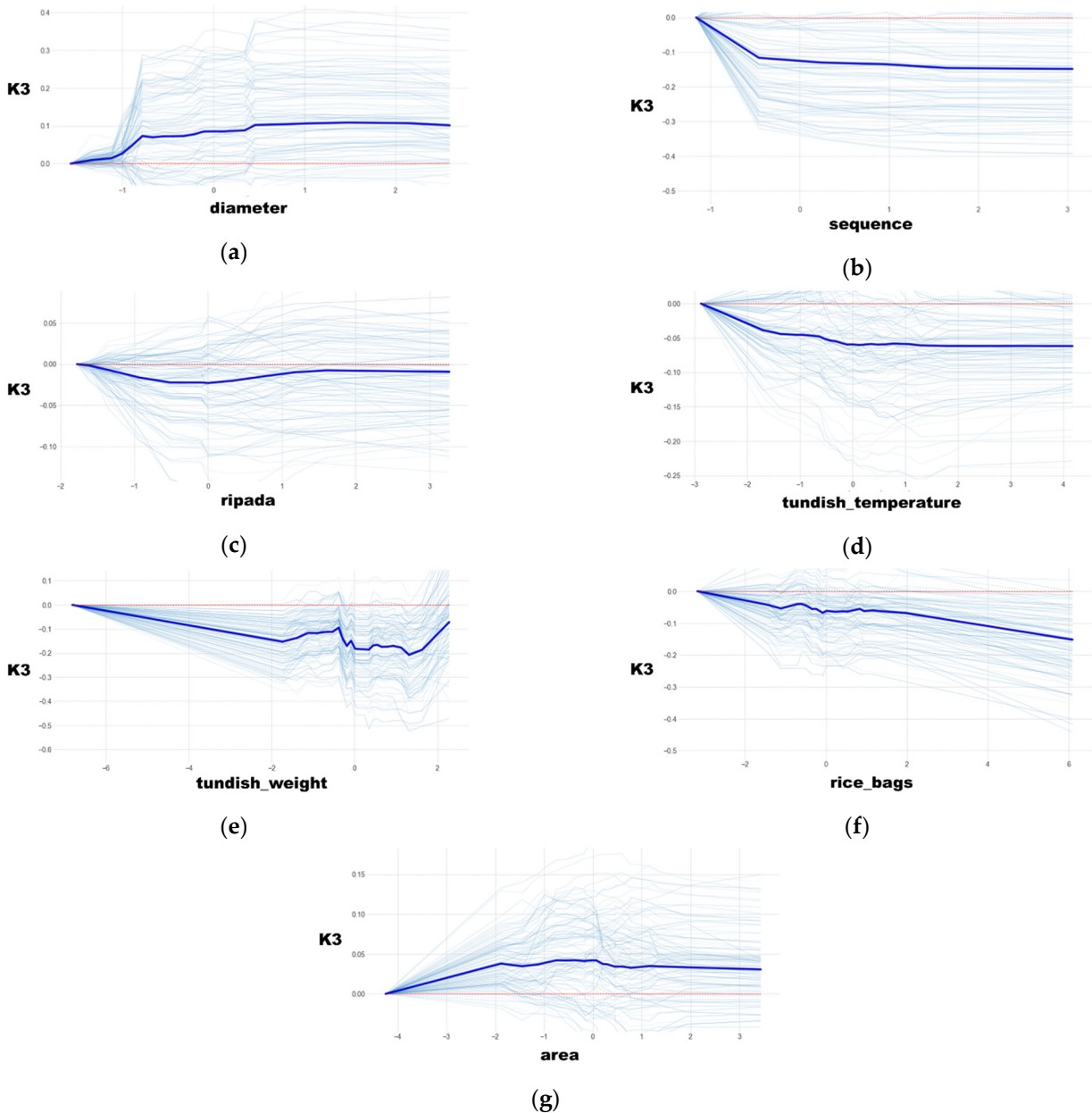

**Figure 6.** PDPs obtained through the RF model on the seven most relevant features identified by means of the PI algorithm: (**a**) diameter, (**b**) sequence, (**c**) ripada, (**d**) tundish temperature, (**e**) tundish weight, (**f**) rice bags and (**g**) area.

## 4. Discussion

This study is aimed at identifying the variables that most influence the K3 index in order to implement corrective measures during the manufacture of steel and improve its inclusionary cleanliness. To achieve an adequate understanding, it is necessary to interpret the results derived from the ML study in terms of the physical processes involved during manufacturing, as well as with the variables that participate in the experimental determination of the K3 index. In the following discussion, the role played by the variables identified in Sections 3.2.2 and 3.2.3 is analyzed and interpreted.

- The variable "diameter", i.e., the final diameter of the rod after rolling, stands out as a very important feature in Figure 5. At first sight, this result is unexpected because it is not truly a manufacturing parameter and it may seem peculiar that the final diameter of the bar influences the inclusion content. However, it is possible to outline two explanations, mechanistically grounded, to understand the possible influence

of the diameter. First of all, it is necessary to consider that the rolling of the steel is a plastic deformation process in which the bar is longitudinally stretched and, simultaneously, transversally shortened. As a matter of fact, the volume of metals remains constant [22,52] during plastic deformation; this is a consequence of the fact that the mechanism justifying plastic deformation at the microstructural level corresponds to the displacement of dislocations [53]. Figure 7 shows a simplistic diagram showing the influence of the rolling process on the length, "L", and diameter, "D", of the bar. Two situations, represented with subscripts 1 and 2, are sketched. The box in dotted lines in Figure 7 represents the area examined with the optical microscope to quantify the number of non-metallic inclusions. The condition of constant volume is expressed in Equation (7). The number of inclusions per unit surface, "n", present in a longitudinal section, is expressed in Equation (8). Introducing the condition of constant volume, it is obtained that n is proportional to the diameter (or, equivalently, inversely proportional to the square root of the length). Therefore, based on this simplified geometric model, there must be a positive correlation between the diameter and the inclusion cleanliness (or, in other words, the K3 index). There is, however, another phenomenon that can contribute to explaining the influence of the diameter. It has been reported, see Vasconcellos et al. [5], that when steels are deformed (during rolling, for example), depending on temperature and inclusion composition, NMIs may deform or crack (or display a mixed behavior). Holappa and Wijk [2] distinguish four types of behaviors: (i) Alumina inclusions, which are hard and brittle, are typically broken up into fragments. (ii) Silicates and manganese sulfides are ductile and deform in a similar way to the steel matrix. (iii) Calcium-aluminate inclusions display very limited deformation and can lead to the formation of cavities after a very demanding rolling. (iv) Complex multiphase inclusions have a hard core surrounded by a deformable phase. Therefore, they show a ductile behavior at low degrees of deformation and prolonged ends at higher deformations. This classification exemplifies the intrinsic difficulties in capturing the complexities of the inclusion content on a simple index. In addition, when an NMI is broken during rolling, the size of the resulting fragments may be lower than the detection limit of the K3 method. Then, the lower the diameter, the smaller the final sizes and, therefore, the lower the value of the K3 index. This rationale is consistent with the results represented in the PDPs; thus, it is observed that the smaller the final diameter, the lower the probability of obtaining a high value of K3. This effect tends to attenuate for large diameters (more than one standard deviation beyond the mean), which also agrees with the argument sketched above.

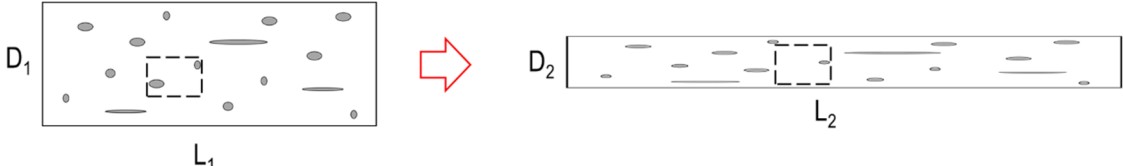

**Figure 7.** Sketch showing the modification in shape and distribution undergone by the NMIs as a consequence of the rolling process.

$$L_1 \cdot (D_1)^2 = L_2 \cdot (D_2)^2 = C \tag{7}$$

$$n = \frac{N}{L \cdot D} = \frac{N}{C} \cdot D \quad \rightarrow \quad \frac{n_2}{n_1} = \frac{D_2}{D_1} = \sqrt{\frac{L_1}{L_2}} \tag{8}$$

- During casting, the content of consecutive LFs is poured into the tundish. The tundish is a small refractory distributer placed over the mold that receives the steel from

the LF [54]. The tundish is in charge of matching the flow rate of liquid steel into the mold with the speed of the strands out of the mold, which is a key aspect of continuous casting. The content of consecutive LFs is poured into the tundish and the feature "sequence" expresses the number of the corresponding LF. Casting is expected to be a continuous process; nevertheless, casting transitions occur at the start of a sequence, during LF exchanges or at the end of casting [55]. NMIs are often generated during transitions [56] because during these non-steady state casting periods, slag entrainment and air absorption are more likely, inducing reoxidation. For example, it has been reported [57] that the presence of slivers (line defects that appear on the surface of the finished product, resulting from NMIs near the surface of the slab) at the beginning of the first heat is five times higher than at the middle of the first heat and over 15 times that for successive heats. Moreover, Zhang and Thomas [55] have shown that the first heat has more total oxygen than the intermediate heats, which facilitates reoxidation. The results represented in Figure 6 reveal a marked influence of the variable sequence in agreement with the previous explanation. Specifically, the probability of belonging to class 1 is larger for the first LFs but it attenuates as sequence increases.

- Feature "billet sequence" has also been identified as relevant. Steel flows from the tundish through a submerged entry nozzle into one or several molds; in this case, the material was fabricated in a six-strand bow-type caster. Therefore, the main function of the tundish is not only to be a steel reservoir between the ladle and the mold, but to distribute the liquid into the six molds (giving rise to six strands/lines). Steel emerges horizontally in the form of a solid steel strand. At this point, it is cut to length using automatic gas burners to produce billets. The index that defines the position of the slab in each strand/line corresponds to the feature billet sequence. As can be seen in Figure 6, best results for billet sequence are obtained when this variable is in the range of mean $\pm$ standard deviation (between $-1$ and $+1$ in the PDP). This result reflects that the inclusion cleanliness of the central billets coming from the same LF is slightly superior to that of the initial and final billets. This matches with the fact that a steady condition leads to better cleanliness than an unsteady one (start and end of the ladle).

- The task of the tundish is of special importance during ladle change and the processes that take place in the tundish deserve some consideration in order to understand the role played by the temperature ("tundish_temperature") and weight ("tundish_weight"). The tundish, like the ladle, is in fact a metallurgical reactor where interactions between the molten steel, the slag, the refractory phases and the atmosphere occur. Regarding temperature, the performance of the tundish is associated with a good thermal insulation of the molten steel, the prevention of reactions with the atmosphere and the absorption of NMIs. Thermal insulation is best achieved with a solid powder layer, while preventing the reaction with the atmosphere and promoting the absorption of nonmetallic inclusions requires the presence of a liquid layer. If the temperature of the tundish is very low, the viscosity of the tundish flux will be too high and its ability to absorb inclusions will be reduced. Conversely, an excessively low viscosity is not recommended because the flux may be drawn down into the mold, producing the contamination of the steel after solidification [58]. The change of ladle makes the process discontinuous. Thus, the temperature of the steel coming from the new ladle is higher than the melt in the tundish from the previous ladle. This temperature difference may affect the flow phenomena in the tundish because convection makes hotter steel ascend in the tundish whereas colder steel descends to the bottom [59]. The influence of the feature tundish_temperature is clearly seen in the corresponding PDP; see Figure 6, where the probability of belonging to the class above the median may be reduced by more than 5% by using above mean values.

- The tundish weight, in turn, acts as a proxy to estimate the steel level in the tundish. During casting operation, some ladle slag can be drawn by vortex formation into the tundish as the metal level in the ladle decreases. Some of this entrained slag may

be carried over into the mold generating defects in the final product. From flotation considerations, there may also be a higher content of inclusions in the last portion of steel to leave the ladle [56]. Moreover, near the end of a ladle, slag may enter the tundish, due in part to the vortex formed in the liquid steel near the ladle exit. This phenomenon requires some steel to be kept in the ladle upon closing [55]. Therefore, it is important to establish the range of steel levels in the tundish that prevent incurring any of these situations. In recent years, numerical modeling has gained importance as a method for determining the fields of fluid flow in the tundish (residence time distribution, velocity profile, temperature distribution, NMIs distribution, etc.) [59]. In general, high residence times and avoidance of short circuits and cold spots of liquid steel are highly appreciated targets and these can be estimated numerically. The PDP in Figure 6 shows that when the tundish weight is very low, the K3 index tends to display larger values, which agrees with the previous rationale.

- The origin of macro- and microinclusions is different. Macroinclusions are typically formed due to reoxidation, ladle slag carryover, slag emulsification, refractory erosion, etc. Microinclusions are typically deoxidation products [59]. The free surface of the melt in the tundish is usually covered by synthetic slag intentionally added, keeping air away from the steel to avoid reoxidation and heat losses from the melt; the feature "slag_bags" represents the number of units used for this purpose. Another important function of the covering slag is the reception of the inclusions from the steel melt. Therefore, it has a key role in the inclusion cleanliness of the final product. Shielding is improved in this process by pouring steel by means of refractory tubes that are immersed in the steel through the slag layer [54]. Emulsification, i.e., the dispersion of slag into the steel as droplets, should be minimized [59]. Rice husk ash (RHA) has historically been the first element to be employed as a tundish covering material due to its availability and low cost. Rice husk is the film that covers rice grains and is commonly used as a biofuel; RHA is the residual of this process, which is formed by silica (~80% wt.) and carbon (5–10% wt.), and is considered an excellent option for its low bulk density and efficiency in preventing heat loss of the molten steel in the tundish [60]. The feature "rice_bags" refers to the amount of RHA employed to thermally isolate the molten material given that, as explained above, an adequate temperature is crucial for the cleanliness of the final product. Figure 6 proves that the higher the number of rice_bags the better the K3 index. Even though this result agrees with the previous argument, it should be treated with caution since, in general, rice bags are usually added at the beginning of each sequence and, as a consequence, there is presumably a correlation between both variables. In fact, this has been observed (the Pearson's correlation coefficient between the features sequence and rice_bags is 0.598). Something similar occurs with the variable slag_bags. Moreover, recent developments [61] suggest that rice husk is harmful to steel cleanliness due to the steel reoxidation by the silica present in rice husk ash.

- The variable "area" represents the size of the surface scanned in each microscopic observation to detect inclusions. Its nominal value is 100 mm$^2$, however, in practice it presents some variability. Given that, as indicated in Section 2.2, the result of the K3 index is normalized (its value is expressed for an area of 1000 mm$^2$), it may be striking that this variable has some influence on the cleanliness. However, a positive correlation is observed in Figure 6, between this feature and the probability of yielding high values of the K3 index. To interpret this result, it is necessary to consider that the last revision of the K3 DIN 50602 [9] test standard was drawn up in 1985 and that, since then, the quality of steels has improved substantially, particularly with regards to their inclusionary content. With the steel analyzed in this study, it is very common to obtain K3 = 0 or a very low value, see Figure 2. Therefore, the "edge effects", sketched in Figure 8, may be playing a role in the value of the K3 index. In this example, a small increase in the inspection area leads to the addition of one

inclusion that otherwise would have been ignored. Logically, this effect diminishes as the inspection area increases.

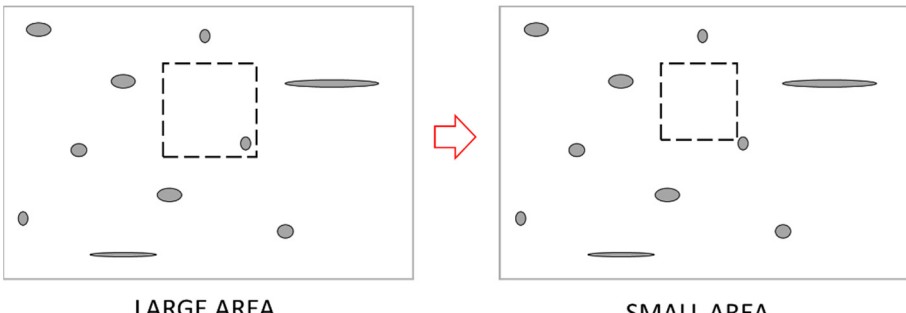

**Figure 8.** Schematic description showing the influence of edge effects on samples with a reduced value of the K3 index.

## 5. Conclusions

Steel fabrication is a suitable environment for the application of Machine Learning methods due to the complexity of the processes that occur during casting. Several classification algorithms have been implemented in this study to predict the K3 index of a clean cold forming steel fabricated by electric arc furnace and rolling. Training, validation and testing were conducted using the experimental results obtained in the context of the quality program of the factory. The dataset consisted of 73 features and 13,616 instances (each one corresponding to the K3 index of a steel coil). The following algorithms have been optimized: Logistic Regression, K-Nearest Neighbors, Decision Tree, Random Forests, AdaBoost, Gradient Boosting, Support Vector Classifier and Artificial Neural Networks (Multi-Layer Perceptron). The best results were achieved by means of Random Forest (AUC = 0.822, *Accuracy* = 0.747, *Precision* = 0.749 and *Recall* = 0.709) which was subsequently used for the assessment of feature importance and correlations.

The Permutation Importance method has been used to identify the manufacturing parameters that most affect the probability of obtaining a K3 index above the median; these can be categorized in two families, namely, manufacturing and non-manufacturing features. Among the former, several variables related to the processes that occur during secondary metallurgy (such as the number of the ladle furnace in the sequence of fabrication or the billet sequence) and tundish operations (tundish weight and temperature) have been identified. The quantitative influence of these variables has been determined by means of Partial Dependence Plots. Based on metallurgical arguments, it has been possible to obtain a coherent picture of the role played by each of these variables enabling the improvement of understanding of the physical processes involved during manufacturing. Thus, a higher inclusionary content has been observed in the steel associated with the first ladle furnace poured into the tundish, which is a consequence of the slag entrainment and increased air absorption that occurs during these non-steady state casting periods. This rationale may also explain the superior cleanliness observed for the central billets coming from the same ladle furnace. Tundish conditions have also been identified as influential in the value of the K3 index. Specifically, the data prove that a low tundish temperature should be avoided; this is explained because lower temperature increases the viscosity of the tundish flux reducing its ability to absorb inclusions. Again, particular attention must be paid to the discontinuous process of ladle exchange due to the non-steady state effects produced. The steel level in the tundish is measured through the weight of the tundish, which is also an important predictor for the K3 index: When the tundish weight is very low, the inclusionary content tends to increase, which is explained due to poor tundish metallurgy performance and also as a consequence of the slag introduced into the tundish as the metal level in the ladle decreases together with the higher content of inclusions in the last portion of steel to leave the ladle.

Two non-manufacturing variables have also been found to play a relevant role in the K3 index. Firstly, a positive correlation was observed between the final diameter of the coil after rolling and the probability of displaying a high value of K3. This surprising outcome has been interpreted considering the changes induced by the process of rolling in the geometric distribution of inclusions as well as the fracture that these may undergo. In addition, the inspection area during the microscopic examination of the samples also displays a positive correlation with the value of the K3 index. A plausible explanation has been proposed, based on the edge effects that may occur during the metallographic examination.

Various results derived from this study are worthy of being implemented in the steel manufacturing process. First, some methodological recommendations: In view of the influence of the non-metallurgical variables on the value of the K3 index, it is recommended to standardize the metallographic process by always using the same area of 100 mm$^2$ during the inspection of samples obtained from wires with the same final diameter after rolling, to avoid introducing additional noise in the dataset. In addition, the Partial Dependence Plots of the metallurgical variables provide the steel manufacturer with a tool to design tailor-made heats with specific values of the relevant features (sequence, tundish weight and temperature, billet sequence, etc.) to optimize the inclusionary cleanliness of the final product. In summary, the use of an analytic procedure based on Machine Learning algorithms has provided useful information for decision-making focusing on the most relevant features of the process, thus facilitating the implementation of corrective measures.

**Author Contributions:** E.R.: conceptualization, methodology, data curation, formal analysis. D.F.: conceptualization, methodology, writing—original draft, writing—review and editing. M.C.: formal analysis, writing—review and editing. L.L.: formal analysis, writing—review and editing. P.M.R.d.Á.: formal analysis, writing—review and editing. A.L.: data curation, writing—review and editing. F.E.: writing—review an editing, funding acquisition. F.G.-S.: conceptualization, methodology, writing. All authors have read and agreed to the published version of the manuscript.

**Funding:** This Project was carried out with the financial grant of the programs I + C = +C 2017 and INNOVA 2018. The financial contribution from SODERCAN, the European Union (program FEDER) and the Government of Cantabria are gratefully acknowledged.

**Data Availability Statement:** Data are not available due to confidential requirements.

**Acknowledgments:** The authors would like to express their gratitude to the technical staff of GLOBAL STEEL WIRE and, especially, to Rafael Piedra and Santiago Pascual, without whom it would not have been possible to carry out this research.

**Conflicts of Interest:** The authors declare no conflict of interest.

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
