# Peer review of "Machine Learning Methods for the Prediction of the Inclusion Content of Clean Steel Fabricated by Electric Arc Furnace and Rolling"

_metals, doi:10.3390/met11060914_

Round 1

Reviewer 1 Report

This is a very interesting and original study on applications of machine learning in inclusions. The methodology is novel and the results are important. This paper should be accepted for publication in metals.

Minor comment:

The discussion on the feature of "rice_bags" should be reconsidered.  According to recent studies, rice husk is harmful to steel cleanliness due to the steel reoxidation by silica in rice husk ash. Please see the following references: 10.1007/s11663-017-0971-3

.

Reviewer 2 Report

This is an interesting study on application of Machine Learning to Steel manufacturing. 

Author Response

Thanks for reviewing the manuscript.

Reviewer 3 Report

The work connects machine learning method and its application for optimisation of impurities presence in steel. It is based on an extremely valuable set of data from the factory, which data were used to optimize the production process in terms of minimizing/optimizing the content of non-metallic impurities. It was prepared extremely carefully, and the authors did not omit most of the additional information. The work is not only a valuable elaboration of a given problem, but even a kind of example of the proper and exhaustive use of machine learning metodes. I especially appreciate the detailed description of the steel production process, the description of all the methods considered (with lot of sources!) and the discussion of the gained results. I am not able to precisely assess the quality of the analyzes carried out because the level of work exceeds my personal experience and skills. However, I am convinced that the volume of the presented data will allow a more experienced reader to better assess the quality of the presented analysis, as the authors do not overlook many details that are irrelevant to less precise authors. The authors do not describe each of the described parameters in detail, but it seems to be a reasonable approach. However, the most significant parameters are also discussed, along with their importance in the context of the analyzed process. It remains for me to congratulate you on your serious approach to the project and suggest one change:

The work lacks a simple K3 index characteristics, without reference to DIN 50602. This should be in place of the first mention of K3 and will certainly improve the already excellent clarity of the literature section.
